# Self-Efficacy, Social Activity, and Spirituality in the Care of Elderly Patients with Polypharmacy in Germany—A Multicentric Cross-Sectional Study within the HoPES3 Trial

**DOI:** 10.3390/healthcare9101312

**Published:** 2021-10-01

**Authors:** Noemi Sturm, Regina Stolz, Friederike Schalhorn, Jan Valentini, Johannes Krisam, Eckhard Frick, Ruth Mächler, Joachim Szecsenyi, Cornelia Straßner

**Affiliations:** 1Department of General Practice and Health Services Research, University Hospital Heidelberg, 69120 Heidelberg, Germany; joachim.szecsenyi@med.uni-heidelberg.de (J.S.); cornelia.strassner@med.uni-heidelberg.de (C.S.); 2Institute of General Practice and Interprofessional Care, University Hospital Tübingen, 72076 Tübingen, Germany; regina.stolz@med.uni-tuebingen.de (R.S.); friederike.schalhorn@med.uni-tuebingen.de (F.S.); jan.valentini@med.uni-tuebingen.de (J.V.); 3Institute of Medical Biometry, University Hospital Heidelberg, 69120 Heidelberg, Germany; krisam@imbi.uni-heidelberg.de; 4Research Center Spiritual Care, Department of Psychosomatic Medicine and Psychotherapy, Technical University of Munich, University Hospital Rechts der Isar, 81675 München, Germany; eckhard.frick@tum.de (E.F.); ruth.maechler@tum.de (R.M.)

**Keywords:** medication management, medication adherence, beliefs about medicines, spirituality, self-efficacy, elderly, multimorbidity

## Abstract

About one third of Europe’s elderly population takes ≥5 drugs. Polypharmacy increases their risk of adverse drug reactions. To ensure drug safety, innovative approaches are needed. The aim of this cross-sectional study was to explore the relationship between psychosocial factors and medication-related beliefs and behaviors. Medication lists of 297 patients were recorded according to the ATC classification. Correlations between the dependent variables, Medication Adherence (MARS) and Beliefs about Medicines (BMQ), and independent variables, General Self-Efficacy (GSE), self-efficacy for managing chronic diseases (SES6G), spiritual needs (SpNQ), patient activity (PAM), loneliness (DJG), and social networks (LSNS), were measured. Patients with higher self-efficacy (OR: 1.113; 95% CI [1.056–1.174]; *p* < 0.001) or self-confidence in managing their chronic condition (OR: 1.188; 95% CI [1.048–1.346]; *p* < 0.007) also showed higher adherence. Lonely patients (OR: 0.420; 95% CI [0.267–0.660]; *p* < 0.001) and those with a need for inner peace (OR: 0.613; 95% CI [0.444–0.846], *p* = 0.003) were more likely nonadherent. Stronger positive beliefs about medications’ usefulness weakly correlated with higher scores on the SES6G (ρ = 0.178, *p* = 0.003) and GSES scale (ρ = 0.121, *p* = 0.042), patient activity (ρ = 0.155, *p* = 0.010) and functioning social networks scale (ρ = 0.159, *p* = 0.008). A weak positive correlation was found between loneliness and the belief that drugs were harmful (ρ = 0.194, *p* = 0.001). Furthermore, interesting correlations were detected regarding the number of medications and overuse beliefs. Psychosocial factors, such as self-efficacy, loneliness, and spiritual needs and medication-related beliefs and behaviors seem to interrelate. Addressing these factors may improve medication management and drug safety.

## 1. Introduction

### 1.1. Prevalence and Risks of Polypharmacy

Medication is used to prevent or treat a disease and the prescription of a drug is often the first therapeutic approach [1]. In contrast, the prescription of multiple medications in patients suffering from more than one chronic condition has become a global healthcare issue [2,3]. Polypharmacy, often described as the regular intake of ≥5 drugs [1], has been linked to multiple negative healthcare outcomes constituting a major challenge for healthcare systems in relation to costs as well as care [4,5].

With increasing life expectancy and a shift in the population pyramid worldwide, the prevalence of chronic diseases rises, and so does the consumption of medications [1,6]. Looking at 17 European countries, the prevalence of polypharmacy ranges from 26.3% to 39.9% among the elderly population [1]. Thereby, both prescription and nonprescription medications add to polypharmacy [1,7]. Their concurrent use is frequently seen in the older generation and involves the danger of many adverse drug events (ADEs), contributing to mortality and morbidity [5,8,9], and at the same time, it may impede the detection of a disease by masking its symptoms [10].

### 1.2. The Role of Medication Nonadherence and Beliefs about Medicines

Oftentimes, a multifaceted consequence of polypharmacy is medication nonadherence [11,12]. In a systematic review and meta-analysis by Mongkhon and Kongkaew [12] a higher prevalence of medication nonadherence is reported in the elderly population with a median of 43.2%, and this again is one reason for ADEs [13].

More than half of the ADEs associated with polypharmacy are considered preventable and caused by inappropriate prescribing, though 20% are linked to low medication adherence [14]. Pasina et al. [15] found that nonadherence was particularly present among those with the highest number of medications. However, it is not only the number of medications that negatively correlates with medication adherence; low health literacy [16,17], information given by package leaflets, different routes of application, timing, and diet restrictions may leave patients feeling overwhelmed by their complex medication regimes, influencing medication-taking behavior [15].

Another aspect that influences medication adherence is patients’ beliefs about medicines [18,19,20], though study outcomes are inconsistent [18,19,21]. In a study by Jäger et al. [22] the authors concluded that minimizing concerns in nonadherent, more worried patients might be as important as raising awareness in those less bothered.

### 1.3. Improving Medication Adherence

According to several systematic reviews [23,24,25,26], interventions that aimed to improve medication adherence showed no or moderate effects. They focused on developing tools to reduce the risk of ADEs or the implementation of structured medication reviews that would allow deprescribing [25,26,27]. Yet, medication management involves more than that. Three components should be considered: 1. structured medication counseling including a brown bag review, 2. the use of regularly updated and comprehensive medication lists comprising both prescription and nonprescription medication, and 3. medication reviews [28].

Although medication counseling is an essential part of medication management, there is a risk of focusing too much on pharmacological information [29] which represents only one part of health literacy. Ostini and Kairuz [16] even concluded that obtaining information alone is not enough when addressing nonadherence. In fact, due to the U-shaped relationship suggested for health literacy and nonadherence, more individualized approaches are required. Nonadherent patients with low health literacy might particularly benefit from interventions strengthening self-efficacy [16]. Furthermore, there is evidence suggesting that when exploring patients’ therapeutic aims and needs, it might be beneficial to not only address pharmacological issues, but to explore psychosocial factors, such as self-efficacy, loneliness, and spirituality, as they might have a direct effect on or a mediating role in medication adherence [30,31,32,33,34]. This might fill the gap that Marcum and Gellad [11] pointed out when concluding that the reasons behind patients’ individual behavior in relation to medication adherence are still not fully understood. Further research is needed to explore the role of psychosocial factors for medication adherence and other medication-related behaviors. This could inform the development of interventions that will change medication management into a more patient-centered and holistic practice, allowing us to increase medication adherence and patients’ health outcomes [11].

### 1.4. Research Questions

Therefore, the aim of this cross-sectional study was to explore the relationships between psychosocial factors and medications, medication-related beliefs as well as behaviors by conducting univariate analyses, asking:

Are there correlations between 1. self-efficacy, 2. loneliness, and 3. spirituality, and

(a)medication adherence,(b)beliefs about medicine, and(c)the number of medications?

## 2. Materials and Methods

### 2.1. Study Design, Setting and Participants

This observational cross-sectional study was carried out on a sample of elderly patients participating in the HoPES3 study—Holistic Care Program for Elderly Patients to Integrate Spiritual Needs, Social Activity and Self-Care into Disease Management in Primary Care [35]—a cluster-randomized trial conducted in Southwestern Germany. The following sections are based on the STROBE reporting recommendations of observational studies [36]. Inclusion criteria were age ≥70 years, suffering from ≥3 chronic conditions, taking ≥3 medications on a long-term basis, participating in at least one Disease Management Program (DMP), being able to take part in a spiritual needs assessment and to complete the questionnaires according to their family physician’s personal estimation, as well as having the ability to give informed consent [35].

### 2.2. Recruitment and Data Collection

Once family practices had given consent to take part in the HoPES3 study, family physicians were asked to recruit eligible patients. Recruitment of patients began in April 2019 and finished in September 2019. For this purpose, patients with DMP appointments in the following three months, who met the inclusion criteria of the study, were extracted from the family practices’ electronic database, and invited to join and contribute to the study. At the upcoming DMP appointment, patients were informed about the HoPES3 study. Those who had decided to take part in the study received a set of questionnaires to fill in for baseline assessment, which was used for correlation analyses in this study (see below). Confirmation of signed consent forms indicating patients’ pseudonyms, DMP enrolment documents, and excerpts from the medical record showing patients’ diagnoses and current medications were sent to the study central office.

### 2.3. Variables

#### 2.3.1. Number of Medications

Patients’ medication lists were used to calculate the number of long-term medications prescribed. Multiple daily dosing and acute medications such as antibiotics were disregarded. Both prescription-only as well as non-prescription medications were recorded in a database and transcribed according to the German Anatomical Therapeutic Chemical (ATC) Classification System [37]. Since one drug could be ascribed more than one ATC code depending on its indication, patients’ corresponding diagnoses were consulted to avoid misallocation. Drug combinations including active ingredients for which separate ATC codes could be found were split to depict the total number of active ingredients.

#### 2.3.2. Sociodemographic Data and Validated Survey Instruments

Next to participants’ sociodemographic data, including age, gender, marital status, personal living situation, level of education, and religious affiliation, data from several validated survey instruments were collected at baseline. On the practice level, information on the type of family practice, family physicians’ work experience and complementary training were documented.

The validated survey instruments served to elicit patients’ views on therapy adherence, attitude towards medication, self-efficacy, spiritual needs, social activity, and loneliness.

##### Medication Adherence Report Scale (MARS-D)

The MARS-D was used to assess patients’ self-reported adherence. It comprises five statements in relation to medication intake and dose adjustment with response options ranging from ‘always’ (scored 1 point) to ‘never’ (scored 5 points). The sum of all five items represents the adherence score, higher scores indicating better adherence [38]. Since the results of the MARS-D were skewed to the right, suggesting full adherence, and to address social desirability bias, this variable was also dichotomized (a score <25 indicating nonadherence; 25 = adherence) as proposed by Fischer et al. [39] who made similar observations.

##### Beliefs about Medicines Questionnaire (BMQ-General)

The Beliefs about Medicines Questionnaire consists of 28 items and six scales [20,40]. In this study, the three subscales of the BMQ-General (BMQ), the *general overuse*, the *general usefulness*, and the *general harm* scale, were used to explore patients’ attitudes and expectations towards medication.

Thereby, *general overuse* investigates the magnitude of the perception that physicians easily trust in medicines, tend to overprescribe, and would prescribe less if they spent more time listening to their patients. *General harm* focuses on the idea of medications being harmful, having toxic and addictive potential, plus suggesting taking a break from medications occasionally. The *general usefulness* subscale assists with the identification of the perception of medications’ advantages and the prospects of leading a better and longer life, but also its alleviating and healing promises. Each subscale comprises four items with a 5-point Likert scale (0 = strongly disagree to 4 = strongly agree). Higher scores (range 4 to 20) represent stronger beliefs. Sum scores were used for statistical analysis [20].

##### General Self-Efficacy Scale (GSE)

The General Self-Efficacy Scale measures the respondent’s optimistic conviction of having the competence to overcome or deal with difficult situations. In particular, it is supposed to predict one’s ability to cope with life [41]. Multiple studies have confirmed the study’s validity [42]. It is a unidimensional scale of ten items with 4 response options (1 = not at all true, 2 = not true, 3 = somewhat true, 4 = very true) indicating whether patients agree or disagree with the items’ statements. The scale score (range 4 to 20) was achieved by summing up the individual scores [41].

##### Self-Efficacy for Managing Chronic Diseases 6-Item Scale (SES6G)

The SES6G assesses patients’ self-confidence in managing their chronic disease despite having to deal with physical or emotional distress. Six questions can be answered on a 10-point Likert scale (1 = not at all confident; 10 = totally confident). The scale score (range 1 to 10) was calculated by the mean of all six items; at minimum four items had to be completed [43]. 

##### Spiritual Needs Questionnaire (SpNQ-20)

The SpNQ-20 allows the measurement of patients’ psychosocial, existential, and spiritual needs. It is a research instrument that can also be used during doctor-patient consultations. *Existential* and *religious needs* are investigated in 6 items. *Needs for inner peace* and *giving needs* are each addressed in four items. In this study, two additional items were included to explore patients’ *family needs*. All items use a 4-point scale to explain the unmet needs’ intensity (0 = no need, 1 = low need, 2 = strong need, 3 = very strong need) [44].

##### Patient Activation Measure 13 (PAM13-D)

The PAM13-D allows quantifying patients’ health-related self-care activities by a sum score of 13 items (1 = strongly disagree, 2 = disagree, 3 = agree, 4 = strongly agree) ranging from 13 to 52. Higher scores indicate higher levels of patient activation [45].

##### Six-Item (Short) De Jong Gierveld Loneliness Scales (DJG 6-Item)

The De Jong Gierveld 6-item loneliness scale allows this study to differentiate between emotional and social subscales. Items of the emotional subscale address negative feelings, such as emptiness or rejection, while items on the social subscale are positively worded, giving a sense of the respondent’s social contacts and ability to trust. Because response categories remain the same, either approving the given statement or disagreeing, before calculating the mean of all six items, responses to the social subscale were reverse coded [46]. The variable was also dichotomized, choosing the same cut-off point as suggested by Huxhold et al. [47], so that scores > 2.5 indicate loneliness, allowing the comparison of this sample to the wider elderly population in Germany.

##### Lubben Social Network Scale (LSNS-6)

The LSNS-6 allows an estimation about the respondent’s social isolation by measuring the number of their social contacts with friends or family members (none = 0, one = 1, two = 2, three or four = 3, five through eight = 4, nine or more = 5) with whom they meet or talk within one month and feel a mutual trust to ask for help or to talk about private matters. The total score is generated by the sum of all six items (range 0 to 30). If the limiting value equals or falls below 12, the respondent is considered at risk of social isolation [48].

### 2.4. Study Size

Twenty-eight family practices were recruited for the HoPES3 study. In total, 323 patients had given consent to participate in the study. Of those, 297 completed the baseline assessment and were therefore included in the intention-to-treat analysis of the HoPES3 study. The same population was chosen for this explorative cross-sectional study.

### 2.5. Statistical Methods

The results of the questionnaires described above were recorded in a database (MS Office Access) and for analysis purposes, transferred to SPSS 26.0. Details of the medication lists were first documented in MS Office Excel. Later data were imported to SPSS 26.0.

Absolute numbers and percentages were reported for categorical variables; the mean, standard deviation, and range were expressed for continuous variables.

Relationships between psychosocial variables (GSE, SES6G, SpNQ-20, PAM13-D, DJG 6-item, LSNS-6) and medication adherence (yes/no, according to MARS-D) were analyzed using univariable binary logistic regression models, calculating odds ratios together with 95% confidence intervals.

For the evaluation of correlations between psychosocial factors as independent variables, excluding the binary DJG loneliness variable, and beliefs about medicine subscale scores as well as number of medications as dependent variables, Spearman correlation coefficients were determined together with 95% confidence intervals.

A nonparametric Mann-Whitney U test was performed to explore the association between loneliness (binary DJG loneliness variable) and the aforementioned ordinal dependent variables.

All analyses were conducted using the SPSS Version 26.0. Missing data were not imputed. All resulting *p*-values are of exploratory nature and need to be interpreted solely descriptively. Thus, no adjustment for multiple testing was conducted. *p*-values smaller than 0.05 are regarded statistically significant.

## 3. Results

### 3.1. Sociodemographic and Health-Related Characteristics of Participants

The mean age of the participants was 78.49. Somewhat more than a half of the study population were female. A third of the patients stated that they live alone. Almost all the participants were Christians. Those belonging to another community explained that they attend independent churches of the Christian faith. Most of the patients had finished secondary school. The mean number of medications taken was 7.64. More than two-thirds of the patients were enrolled in the DMP for type II diabetes. Almost 20% of the patients were registered for more than one DMP. More details on patients’ sociodemographic data as well as health and medication-related characteristics are displayed in Table 1.

At the practice level, the mean work experience was 22.37 years. More than half of the participating family physicians were working in single-handed practices (14/27; 51.9%). Thirteen of 27 (48.1%) family physicians, i.e., almost half of the participating family physicians were trained in complementary medicine.

The average scores obtained through the validated instruments are provided in Table 2.

### 3.2. Associations between Psychosocial Factors and Medication Adherence

Patients who reported higher levels of self-efficacy (OR: 1.113, 95% CI [1.056; 1.174], *p* < 0.001) or self-confidence in managing their chronic condition (OR: 1.188, 95% CI [1.048; 1.346], *p* = 0.007), i.e., higher sum and mean scores, respectively, also showed higher adherence (see Table 3). A higher chance of showing medication adherence was also detected in more active patients with higher sum scores on the PAM scale (OR: 1.075, 95% CI [1.028; 1.124], *p* = 0.002).

Lonely patients, those with a mean score > 2.5, (OR: 0.420, 95% CI [0.267; 0.660], *p* < 0.001), and those with unmet needs for inner peace (OR: 0.613, 95% CI [0.444; 0.846], *p* = 0.003) were more likely nonadherent. This was not observed when using the binary variable of the DJG scale nor for any other spiritual need.

### 3.3. Associations between Psychosocial Factors and Beliefs about Medications

Higher scores on the BMQ general usefulness scale, indicating stronger beliefs in medications’ usefulness, weakly correlated with higher scores, i.e., stronger expressions concerning patients’ self-confidence in managing their chronic condition (ρ = 0.178, 95% CI [0.06; 0.29], *p* = 0.003), self-efficacy (ρ = 0.121 95% CI [0.00; 0.23], *p* = 0.042), and patient activity (ρ = 0.155, 95% CI [0.04; 0.27], *p* = 0.010). Conversely, as shown in Table 4, patients with the opposite characteristics expected medications to be more harmful.

Larger social networks (ρ = 0.159, 95% CI [0.04; 0.27], *p* = 0.008) also weakly correlated with the belief medications were useful. Correspondingly, patients with lower DJG mean scores, i.e., those feeling less lonely, (ρ = −0.240, 95% CI [−0.35; −0.13], *p* < 0.001) were associated with stronger beliefs regarding medications’ usefulness.

Spiritual needs were not found to correlate with beliefs about medications’ usefulness.

A weak positive correlation was found between loneliness and the belief that drugs were harmful (ρ = 0.194, 95% CI [0.08; 0.31], *p* = 0.001).

A higher degree of loneliness, i.e., higher scores on the DJG mean and DJG binary scales, showed a weak positive correlation in relation to the belief that medications were overused (Table 4, Figure 1). A similar finding was observed in patients with higher scores on family needs, indicating stronger needs, though the result was not significant. In fact, the Mann-Whitney U test which investigated whether there are differences between lonely and non-lonely patients and their beliefs about medicines was only statistically significant regarding the belief medications were overused, but not for the other two variables *BMQ general usefulness* and *BMQ general harms*. Results were also not statistically significant in relation to the number of medications.

### 3.4. Associations between Psychosocial Factors and Number of Medications

Lower scores on both self-confidence in dealing with chronic diseases (ρ = −0.322, 95% CI [−0.42; −0.21], *p* < 0.001) and the patient activity measure scale (ρ = −0.126, 95% CI [−0.24; −0.01], *p* = 0.035) significantly correlated with an increasing number of medications (see Table 5). Loneliness and spiritual needs did not correlate with the number of medications patients were taking.

## 4. Discussion

The aim of this study was to explore the relationships between psychosocial factors, namely self-efficacy, loneliness, and spirituality, and medication-related beliefs and behaviors, as well as the number of medications taken according to patients’ medication lists. In the following, possible implications of the results for medication management in practice are discussed.

### 4.1. The Role of Self-Efficacy for Medication Management

Interestingly, both a stronger sense of general and health-related self-efficacy was associated with a more positive attitude towards medication and medication adherence. This supports the findings by Martos-Méndez [33] who concluded that nonadherence was more likely in patients with lower self-efficacy, whereas higher perceived self-efficacy was associated with better adherence to either medication or any other health advice. Further, it coincides with Ostini and Kairuz’s [16] proposal to strengthen self-efficacy in those with low health literacy in order to improve their adherence, but this would need to be verified for this sample, as the different levels of health literacy were not investigated.

In turn, one can assume that lower self-efficacy increases the likelihood of polypharmacy. This is because lower self-efficacy has been linked to a greater illness burden and hence a lower quality of life, which then again makes medication use more likely [50]. This corresponds to the findings in this study. Patients who were more confident in managing their chronic condition were taking fewer medications. At the same time, they also appreciated medications’ usefulness. A more negative attitude towards medication could lead to patients not adhering to the intake rules and therefore not experiencing any improvement in their condition. Physicians may then prescribe even more medications contributing to patients’ polypharmacy [51].

Since all three variables indicating self-efficacy significantly correlated with medication adherence, the results might hint that self-efficacy encourages medication adherence. Self-efficacy might also foster a more positive attitude concerning medications’ beneficial properties, but this would need to be confirmed by a longitudinal study design. However, addressing patients’ self-care abilities in the sense of a more holistic medication management might invite patients to become more engaged in their own treatment as they also feel empowered and partly responsible for their treatment outcome. This could, e.g., be achieved by not only focusing on improving patients’ management skills regarding their pharmaceutical therapy but, if suitable, offering non-pharmaceutical options that allow patients to help themselves [52].

### 4.2. Loneliness in Relation to Medication-Related Beliefs and Behavior and Number of Medications

Further, analyses showed that medication adherence declined the lonelier patients felt. This is in line with the results of previous research [53,54,55] that focused on specific chronic conditions. Mondesir et al. [31] found a small association between medication adherence and frequent contact with either family members or friends in patients with risk factors for coronary heart disease. In line with that, Lu et al. [32] reported negative correlations for both social isolation and loneliness in relation to medication adherence. Social support itself has been associated with patients’ self-efficacy. Those with a higher sense of self-efficacy were more likely to be adherent and there is evidence that interpersonal relationships providing social support changes patients’ perception of their own abilities [33]. Park et al. [34] emphasized not to underestimate the impact of these psychosocial factors that promote behavioral change and encourage the self-management of chronic diseases.

It can be assumed that loneliness may be mediated by a lack of social support, especially regarding practical assistance [56,57]. Therefore, Eriksen et al. [56] recommended to involve patients’ social networks as they might facilitate medication adherence by strengthening patients’ self-efficacy. In fact, Hacihasanoglu Asilar et al. [54] found that with increased social support, patients’ perceived medication adherence increased too, whereas loneliness decreased.

Given the results, one might also hypothesize that loneliness may nurture a more skeptical attitude towards medication and negatively affect medication adherence. Being able to identify those patients who feel lonely could be beneficial for both the patient and the prescriber, as there is evidence that loneliness does not only influence health outcomes, but it might also be a direct or indirect result of patients’ current treatment or condition [54]. When reviewing patients’ medications, one could consider that lonely patients might grow lonely because of the medication they require, e.g., due to the type of medication or its application that limits spontaneity and free time to spend with others, thereby increasing the risk of nonadherence [56,58].

Although further evidence is required to explain a causal relationship, one should consider exploring patients’ feelings of loneliness as part of medication management. As elderly multimorbid patients tend to consult their family physicians more often, family physicians could build rapport, allowing them to explore their patients’ needs. However, as a study by Duncan et al. [59] revealed, a lot of times medication reviews are not conducted in the presence of the patient concerned because they are considered too time consuming in daily practice. Moreover, family physicians find it particularly challenging to address patients’ feelings of loneliness, and conversely patients do not easily talk about such personal feelings. In this respect, family physicians should take advantage of training offers to improve their communication skills [60].

### 4.3. The Role of Spirituality in Medication Management

Not only lonely patients, but also those with an unmet need for inner peace were more likely nonadherent. Due to this finding, it can be hypothesized that medication adherence would improve if the need for inner peace was fulfilled. Badanta-Romero et al. [61] found that spirituality as a coping strategy to deal with one’s chronic disease may have a positive as well as negative impact in relation to medication adherence, but stressed that these aspects should be considered in holistic healthcare management.

Just as loneliness has a mediating effect on medication adherence, so might patient’s unmet spiritual needs affect medication adherence, but this would need to be confirmed by further research.

### 4.4. Strengths and Limitations

Due to the nature of the statistical analyses conducted and the cross-sectional study design, no causal relation can be detected, nor are these findings generalizable to a wider population including all age groups. Regarding patients’ medication lists, data might not reflect patients’ actual number of medications. This is because medication lists often remain incomplete. However, since nonprescription medications that were documented on the patients’ medication lists were included in the number of medications presented in this study, the results provide a good estimate of this sample’s mean intake. In addition, although the questionnaire-based research approach relies on patients’ honesty so that recall bias as well as social desirability bias with underestimation or overestimation in relation to sensitive topics such as loneliness cannot be ruled out, data quality can be assumed to be of good value considering the study’s relatively large sample size.

## 5. Conclusions

Not only do polypharmacy and medication adherence interrelate, but they also have several influencing factors in common that should be recognized in medication management. As such, psychosocial aspects that affect patients’ medication-related beliefs and behaviors should not only be considered when planning interventions to improve medication adherence. Preferably, physicians should include these before adjusting medication regimes to elderly patients’ needs. In this context, it might be sensible to explore patients’ feelings of loneliness and possibilities to increase their self-care abilities. Further research is needed to clarify the role of spirituality in relation to medication adherence.

## Figures and Tables

**Figure 1 healthcare-09-01312-f001:**
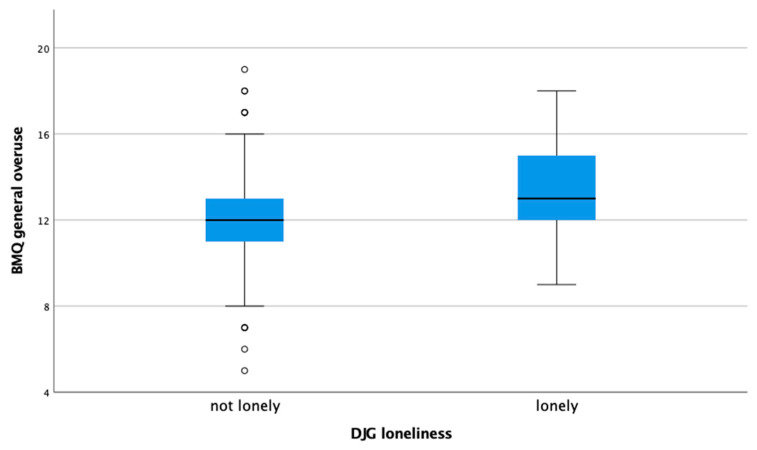
Relationship between loneliness and the perceived belief medication were overused.

**Table 1 healthcare-09-01312-t001:** Main characteristics of study population at baseline assessment.

Main Characteristics	Total *n* = 297	*n* (%) or Mean (Range, SD)
Mean age in years (range, SD)	297	78.46 (70–91; 4.76)
Female *n* (%)	297	163 (54.9%)
Marital status *n* (%)	289	
- Single		101 (34.9%)
- Partnership		188 (65.1%)
Living alone *n* (%)	287	90 (31.4%)
Level of education *n* (%)		
- Primary and secondary* school education (* including both types of German secondary school education Haupt- and Realschulabschluss)		244 (85.1%)
- High school (Abitur—German final school exams)	287	13 (4.5%)
- University degree		26 (9.1%)
Religion *n* (%)	286	
- Christian		248 (86.7%)
- Other		11 (3.8%)
- No religion		27 (9.4%)
Type of Disease Management Program (DMP) *n* (%)		
- DMP for COPD		26 (8.8%)
- DMP for Asthma		21 (7.1%)
- DMP for Diabetes type I	297	2 (0.7%)
- DMP for Diabetes type II		213 (71.7%)
- DMP for Coronary artery disease		99 (33.3%)
Number of DMP registrations *n* (%)	297	
- Registration for only one DMP		235 (79.1%)
- Registration for two DMPs		58 (19.5%)
- Registration for three DMPs		3 (1.0%)
Mean number of medications (range, SD)	297	7.64 (3–20; 2.99)

**Table 2 healthcare-09-01312-t002:** Descriptive results of the survey instruments distributed to the study population at baseline assessment.

Independent Variables *	Total *n* = 297	Mean (Range; SD)
Self-efficacy for managing chronic diseases (SES6G)	290	7.15 (1.0–10.0; 2.06)
General self-efficacy scale (GSES)	278	31.17 (16.0–40.0; 5.20)
Short form health survey (SF12)		
- physical health	263	38.04 (13.2–60.6; 9.88)
- mental health	263	49.75 (21.7–71.1; 10.35)
Lubben social networks scale—social isolation (LSNS-6)	281	16.91 (4.0–30.0; 5.48)
Loneliness (DJG)	290	1.77 (1.00–3.67; 0.61)
Patient activation measure (PAM)	281	43.01 (24.0–52.0; 5.95)
Medication adherence scale (MARS)	292	23.61 (17.0–25.0; 1.68)
Beliefs about Medicines (BMQ)		
- general overuse	288	12.20 (5.0–19.0; 2.33)
- general harms	283	9.34 (4.0–17.0; 2.39)
- general usefulness	291	15.79 (10.0–20.0; 2.18)
Spiritual needs questionnaire (SpNQ)		
- religious needs	233	0.94 (0.0–3.0; 0.91)
- existential needs	225	1.12 (0.0–3.0; 0.74)
- needs for inner peace	242	1.54 (0.0–3.0; 1.50)
- giving needs	234	1.50 (0.0–3.0; 0.82)
- family needs	268	1.90 (0.0–3.0; 0.90)
Independent Variables (binary)	*n* (%)	
Loneliness (DJG)		
- not lonely	259 (89.0%)	
- lonely	32 (11.0%)	
Medication adherence report scale (MARS)		
- nonadherence	187 (64.0%)	
- adherence	105 (36.0%)	

* SES6G = health related self-efficacy scale (mean over all six items, scale from 1 to 10, higher scores indicating greater self-confidence in their own abilities) [43]. GSES = General Self-Efficacy Scale (GSES) (sum score of 4 items, scale from 10 to 40, higher scores indicating a higher level of self-efficacy) [41]. SF12 = Health-related quality of life with scales ranging from 0 to 100, higher scores indicating better quality of life [49]. LSNS-6 = Lubben Social Network Scale (6 item version, sum score ranging from 0 to 30, a score <12 indicating risk of social isolation) [48]. DJGS-6 = De-Jong-Gierveld Loneliness Scale (6-item short version with a 4-point response option, a score <2, 5 indicating loneliness) [46,47]. PAM = Patient Activation Measure (sum of 13 items, total score ranges from 13–52, higher scores indicating a higher level of patient activation) [45]. MARS = Medication Adherence Report Scale (sum of all 5 items, ranging from 5 to 25, higher scores indicating better medication adherence) [38,39]. BMQ = first part of the Beliefs About Medicines Questionnaire addressing general expectations regarding drug treatment in three scales with sum scores ranging from 4 to 20, higher scores confirming the beliefs of medications’ usefulness, overuse, and harmfulness [40]. SpNQ = Spiritual Needs Questionnaire quantifies the strength of unmet needs including psychosocial, existential, and spiritual needs (score ranges from 0 to 3 indicating no need whereas 3 reflects an intense need) [44].

**Table 3 healthcare-09-01312-t003:** Correlations between psychosocial variables and medication adherence.

Independent Variable *	Odds Ratio with 95% CI	*p*-Value
General Self-efficacy Scale (GSES)	1.113 [1.056; 1.174]	<0.001
Self-efficacy for managing chronic diseases (SES6G)	1.188 [1.048; 1.346]	0.007
Patient activation measure (PAM)	1.075 [1.028; 1.124]	0.002
Loneliness (DJG binary)	0.854 [0.386; 1.891]	0.697
Loneliness (DJG)	0.420 [0.267; 0.660]	<0.001
Lubben social networks scale—social isolation (LSNS-6)	1.028 [0.982; 1.075]	0.237
Spiritual needs questionnaire (SpNQ)		
- religious needs	1.076 [0.802; 1.445]	0.625
- existential needs	0.811 [0.555; 1.187]	0.282
- needs for inner peace	0.613 [0.444; 0.846]	0.003
- giving needs	0.937 [0.675; 1.300]	0.697
- family needs	0.881 [0.668; 1.163]	0.372

* A detailed legend can be found below Table 2.

**Table 4 healthcare-09-01312-t004:** Correlations between psychosocial variables and the Beliefs about Medicines (BMQ) general subscales.

Independent Variable	*N*	rho	CI 95%	*p*-Value
BMQ general overuse				
General Self-efficacy Scale (GSES)	271	−0.110	[−0.23; 0.01]	0.070
Self-efficacy for managing chronic diseases (SES6G)	281	−0.084	[−0.20; 0.03]	0.161
Patient activation measure (PAM)	274	−0.071	[−0.19; 0.05]	0.244
Loneliness (DJG)	284	0.203	[0.09; 0.31]	0.001
Lubben social networks scale—social isolation (LSNS-6)	273	−0.016	[−0.13; 0.10]	0.798
Spiritual needs questionnaire (SpNQ)				
- religious needs	226	0.020	[−0.11; 0.15]	0.764
- existential needs	218	0.029	[−0.18; 0.09]	0.519
- needs for inner peace	236	−0.044	[−0.10; 0.16]	0.656
- giving needs	227	0.012	[−0.12; 0.14]	0.858
- family needs	260	0.122	[0.00; 0.24]	0.050
BMQ general usefulness				
General Self-efficacy Scale (GSES)	272	0.178	[0.06; 0.29]	0.003
Self-efficacy for managing chronic diseases (SES6G)	284	0.121	[0.00; 0.23]	0.042
Patient activation measure (PAM)	276	0.155	[0.04; 0.27]	0.010
Loneliness (DJG)	285	−0.240	[−0.35; −0.13]	<0.001
Lubben social networks scale—social isolation (LSNS-6)	275	0.159	[0.04; 0.27]	0.008
Spiritual needs questionnaire (SpNQ)				
- religious needs	227	0.032	[−0.10; 0.16]	0.632
- existential needs	220	0.067	[−0.07; 0.20]	0.325
- needs for inner peace	237	−0.084	[−0.21; 0.04]	0.197
- giving needs	228	0.047	[−0.08; 0.18]	0.477
- family needs	262	−0.097	[−0.22; 0.02]	0.119
BMQ general harms				
General Self-efficacy Scale (GSES)	267	−0.200	[−0.31; −0.08]	0.001
Self-efficacy for managing chronic diseases (SES6G)	276	−0.251	[−0.36; −0.14]	<0.001
Patient activation measure (PAM)	272	−0.159	[−0.27; −0.04]	0.009
Loneliness (DJG)	279	0.194	[0.08; 0.31]	0.001
Lubben social networks scale—social isolation (LSNS-6)	269	−0.096	[−0.21; 0.02]	0.117
Spiritual needs questionnaire (SpNQ)				
- religious needs	224	−0.046	[−0.18; 0.09]	0.493
- existential needs	215	−0.052	[−0.18; 0.08]	0.450
- needs for inner peace	233	0.040	[0.44; 0.63]	0.543
- giving needs	224	−0.024	[−0.15; 0.11]	0.724
- family needs	256	0.086	[−0.04; 0.21]	0.168

**Table 5 healthcare-09-01312-t005:** Correlation analyses between psychosocial variables and number of medications.

Independent Variable	*N*	rho	CI 95%	*p*-Value
General Self-efficacy Scale (GSES)	278	−0.105	−0.22; 0.01	0.079
Self-efficacy for managing chronic diseases (SES6G)	290	−0.322	−0.42; −0.21	<0.001
Patient activation measure (PAM)	281	−0.126	−0.24; −0.01	0.035
Loneliness (DJG)	290	0.060	−0.06; −0.17	0.310
Lubben social networks scale—social isolation (LSNS-6)	281	−0.074	−0.19; 0.4	0.217
Spiritual needs questionnaire (SpNQ)				
- religious needs	233	−0.027	−0.16; 0.10	0.680
- existential needs	225	−0.013	−0.14; 0.12	0.845
- needs for inner peace	242	−0.018	−0.14; 0.11	0.782
- giving needs	234	0.056	−0.07; 0.18	0.394
- family needs	268	0.113	0.23; −0.01	0.066

## Data Availability

The dataset will be made available upon reasonable request.

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
