# Peer review of "Self-Efficacy, Social Activity, and Spirituality in the Care of Elderly Patients with Polypharmacy in Germany—A Multicentric Cross-Sectional Study within the HoPES3 Trial"

_healthcare, 2021, doi:10.3390/healthcare9101312_

Round 1

Reviewer 1 Report

Review of manuscript entilted: “Self-efficacy, social activity, and spirituality in the care of elderly patients with polypharmacy – a multicentric cross-sectional study within the HoPES3 trial” authored by Noemi Sturm, Regina Stolz, Friederike Schalhorn, Jan Valentini, Johannes Krisam, Eckhard Frick, Ruth Mächler, Joachim Szecsenyi and Cornelia Straßner

Thank you for opportunity to review this interesting manuscript.

In the presented manuscript authors analyzed social and spiritual parameters as well as self-efficacy of polypharmacy among elderly patients in Germany. Introduction provides sufficient information about undertaken problem. Methods are described with details. Results are presented clearly, however I have one major concern about them. Discussion and conclusions are supported by the obtained results.

Major concerns:

  • Please provide brief explanation for missing observations in tables, what I mean is that total n was equal to 297, however for example in the mean age in years we see that n = 292. If the age was not known should this observation be included in your analysis?

Minor concerns:

  • Title – I would suggest to mention that study is performed among German people.
  • Abstract – statistical significance values are not necessary here
  • Please consider adding “honesty of patients” in limitation subparagraph, since it is the main limitation of all questionnaire-based studies

Author Response

Dear Sir or Madam. Thank you for your thorough review of our manuscript. Please find our responses to your comments in the attachment. 

Reviewer 2 Report

I would add definitely in the introduction, that health literacy is also a big problem in many countries among elderly. Preparation of tools, for patient counselling in mentioned age group, would be adviced. Example: https://www.frontiersin.org/articles/10.3389/fphar.2021.582200/full please have a read. It would be worth considering in reference and providing a small explanation on this element, maybe you could adres the health literacy in your population in the future.  Mention also health literacy in the discusion. You must have missed that part. I am looking forward for receiving a new version. 

Author Response

(The authors gave the same response as above.)

Reviewer 3 Report

Thank you for the opportunity to review this study on the self-efficacy, social activity, and spirituality in the care of older patients with polypharmacy. This research identifies some interesting facts and presents some considerations to those findings.

General comments:

To improve reader understanding, as all the scores presented are not interpreted in the same way (high levels vs. low levels), I would suggest that you recall each time which clinical result is associated with a high or low score.

Abstract:

  1. Lines 17-34: I would suggest to present the p-values for all statistics in the abstract.

Introduction

  1. General comments: The introduction seems too long and too detailed. I would suggest to reduce it and to better justify the originality of your work. Moreover, some elements should be in the Discussion section.
  2. Line 56: The more appropriate term is "potentially inappropriate prescribing" not “potential inappropriate prescribing”.
  3. Lines 136-143: Could you rephrase the objective sentence to select only one of the objectives as primary objective and define the others as secondary objectives?

Methods:

  1. Could you please indicate that your study is observational?
  2. Could you verify that your study follows the STROBE guidelines and mention it in the method section?
  3. Lines 162-163: You mentioned that the data was declarative so missing data were expected. How do you deal with this issue?
  4. Lines 183-244: Why did you choose to dichotomize some variables and not others? Could you justify this approach in the Method section?
  5. Lines 183-244: Could you provide in the supplementary material the MARS-D, BMQ-General and other questionnaires and examples of score calculation?
  6. Lines 246-247: How did you deal with the hierarchical structure (family practices/patients) of your data? Do you think it would be useful to use mixed models?
  7. Lines 256-270: You only performed univariate analyses (univariable binary logistic regression models, Spearman correlation coefficients, Mann Whitney U test) to assess the associations between psychosocial factors and the variables of interest (medication adherence, beliefs about medicines and polypharmacy). However, there are potential confounding factors, so why not perform multivariate analyses in addition?

Results and Tables:

  1. Lines 273-280 and Table 1: Could you give more information about patients' comorbidities?
  2. Line 273-280 and Table 1: Could you give information on the characteristics of the family practices?
  3. Tables 2 and 3: Could you specify in the legend of these 2 tables the interpretation of the different scales used?
  4. Table 5: This table does not seem necessary and could be replaced by text.

Discussion and Conclusion:

  1. Lines 407-414: Could you add in the limitations section, a sentence about the limits of cross-sectional design?

Author Response

(The authors gave the same response as above.)

Round 2

Reviewer 3 Report

No more comments